# Contrasting Evolutionary Dynamics and Global Dissemination of the DNA-A and DNA-B Components of Watermelon Chlorotic Stunt Virus

**DOI:** 10.3390/v17121571

**Published:** 2025-11-30

**Authors:** Zafar Iqbal

**Affiliations:** Central Laboratories, King Faisal University, Al-Ahsa P.O. Box 31982, Saudi Arabia; zafar@kfu.edu.sa; Tel.: +966-580-776-536

**Keywords:** watermelon chlorotic stunt virus, begomovirus, genetic diversity, phylogeography, recombination, mutation, *Bemisia tabaci*, phylogenetic analysis

## Abstract

Watermelon chlorotic stunt virus (WmCSV), a bipartite begomovirus, poses a severe and expanding threat to global cucurbit and watermelon production, driving an urgent need to unravel its genetic diversity indices and evolutionary complexities. To elucidate its evolutionary history, this study investigated the genetic diversity, evolutionary dynamics, and global dispersal of its genomic components, DNA-A (WmA) and DNA-B (WmB). The analyses uncovered striking contrasts between the components. WmB exhibited markedly greater genetic diversity (π = 0.0508 vs. 0.0119 for WmA), a slightly faster evolutionary rate (1.26 × 10^−4^ vs. 1.44 × 10^−3^ substitutions/site/year), and a far more complex recombination profile, with 34 events detected compared to only one in WmA. The abundance of recombination breakpoints in WmB underscores its central role in shaping genomic variability and adaptive potential. Phylogenetic analyses of both components unveil eight robustly supported clades per segment, predominantly shaped by geographical boundaries, hinting at localized evolutionary trajectories with constrained long-distance gene flow, with the exception of Oman. Bayesian time-scaled phylogenies and phylogeographic reconstructions further illuminate distinct dissemination pathways, suggesting an intriguing origin, with WmA likely emerging from the United States and WmB tracing back to Saudi Arabia, while the Middle East emerges as a dynamic epicenter for regional spread and subsequent incursions into the Americas. Together, these findings reveal contrasting evolutionary forces driving WmCSV diversification and provide critical insights into its origins and ongoing global emergence.

## 1. Introduction

Comparative phylogeographic and genetic diversity studies offer critical insights into how historical and ecological forces shape the genetic structure of widely distributed plant viruses [1]. Patterns of congruent phylogeographic structure across viral lineages often reveal shared evolutionary histories, shaped by common biogeographic barriers, host shifts, and transmission dynamics [2]. While virus-specific factors play a role, widespread concordant patterns frequently point to broader historical influences, including climatic fluctuations, agricultural practices, and human-mediated dispersal [3], which collectively determine the present-day genetic diversity and distribution of plant pathogens [4,5].

Watermelon chlorotic stunt virus (WmCSV), a bipartite begomovirus (family *Geminiviridae*) transmitted by the whitefly (*Bemisia tabaci*), carries two single-stranded DNA components, DNA-A (WmA) and DNA-B (WmB), each approximately 2.6–2.8 kb in size. WmA encodes essential proteins for viral replication, transcription, pathogenesis, and encapsidation, including the replication-associated protein (Rep), pre-coat protein (AV2), and coat protein (CP), while WmB facilitates systemic movement within the host through proteins such as the movement protein (MP) and nuclear shuttle protein (NSP) [6,7]. First reported in Yemen in the late 1980s as a major pathogen of watermelon (*Citrullus lanatus*) [8], WmCSV rapidly spread across the Middle East and North Africa, with detections in Israel, Jordan, Lebanon, Oman, and Palestine [9,10,11,12,13]. It was identified in Saudi Arabia in 2014 [14], and later characterized in zucchini and watermelon [15,16,17]. The virus has since expanded globally, reaching Mexico in 2018 [18] and the United States in 2021 [19,20], with additional reports from Sudan, Iran, and Oman causing major cucurbit losses [21,22]. WmCSV frequently co-occurs with other begomoviruses in mixed infections, complicating disease management [19,22,23].

Infecting watermelon, melon, cucumber, and squash, WmCSV induces chlorotic stunting, leaf curling, and poor fruit quality [24]. It has become a production-limiting factor in the Middle East, North Africa, and South Asia, with its rapid dissemination driven by whitefly migration, climate change, and intensive cucurbit farming [25]. As a major threat to global cucurbit and watermelon production, WmCSV undermines food security. Its genetic diversity—particularly in Rep, CP, and movement proteins—facilitates adaptation to new hosts, regions, and environments [26,27]. High mutation rates, large population sizes, and rapid replication generate diverse genomic variants capable of overcoming selective pressures [28]. Single nucleotide polymorphisms (SNPs), both synonymous and nonsynonymous, further shape gene expression, infectivity, and viral fitness [29]. Such genomic insights are vital for pinpointing recombination hotspots, designing diagnostics, breeding resistant cultivars, and building predictive epidemiological models.

Historically, evolutionary studies of plant viruses focused on RNA viruses due to their high mutation rates, driven by error-prone RNA-dependent RNA polymerases (RdRp) [30]. However, ssDNA begomoviruses exhibit comparable mutation rates to RNA viruses [31,32,33] due to low-fidelity DNA polymerase activity and spontaneous biochemical reactions (e.g., deamination, oxidation) [34,35]. While mutations drive geminivirus diversification [27,36,37], recombination events significantly enhance their evolution and genetic variation [38,39]. Molecular approaches—nucleotide diversity analysis, phylogenetics, and recombination detection—have elucidated begomovirus dynamics, revealing gene flow, bottlenecks, and selection pressures. Yet, despite their global spread and mounting agricultural impact, regional studies on begomovirus diversity remain limited, constraining efforts to predict outbreaks, track virulent strains, and design effective control strategies. Deeper insights into their evolution and population structure are critical for safeguarding crop production and ensuring agricultural resilience.

Despite the growing threat posed by WmCSV, its genetic diversity remains insufficiently characterized in many critical regions, hampering the development of accurate diagnostics and predictive models for emerging strains. Accelerating agricultural globalization, coupled with climate-driven shifts in vector dynamics, has heightened the risk of widespread and more virulent outbreaks, emphasizing the urgent need to unravel the evolutionary forces shaping WmCSV populations. In this study, we present a comprehensive global analysis of WmCSV genetic diversity, shedding light on its population structure, recombination dynamics, and adaptive strategies. By addressing these knowledge gaps, our findings provide a foundation for the development of targeted disease management strategies—including molecular surveillance systems and the breeding of resistant cultivars—thereby enhancing the resilience and sustainability of watermelon production in virus-endemic regions.

## 2. Materials and Methods

### 2.1. Sequence Retrieval and Multiple Sequence Alignment

To investigate the genetic diversity of WmCSV, complete genome sequences for WmA and WmB components were retrieved from NCBI GenBank (https://www.ncbi.nlm.nih.gov/) on 5 April 2025. Accession numbers of all these sequences are listed (Appendix A). From these sequences, nucleotide datasets were created for the full-length WmA and WmB components. Further sub-datasets were generated, isolating each individual open reading frame (ORF). For geographically focused analyses, sequences were also categorized by country of origin, creating distinct subsets for both the full components and their respective ORFs. Multiple sequence alignments (MSA) were then performed on all generated datasets using the Muscle algorithm [40] within the MEGA11 software package [41]. Finally, these alignments were carefully inspected and manually adjusted to maximize accuracy.

### 2.2. Recombination Analysis

Potential recombination events within the WmA and WmB genomes were identified using two complementary approaches: the Recombination Detection Program v5.5 (RDP5) [42] and the Genetic Algorithm for Recombination Detection (GARD) (https://www.datamonkey.org/gard; accessed on 26 April 2025). For RDP analysis, seven algorithms—RDP, BOOTSCAN, GENECONV, MAXCHI, CHIMAERA, SISCAN, and 3SEQ—were applied with default settings, using a Bonferroni-corrected *p*-value threshold of 0.05 to control for false positives. Only recombination events supported by at least five of the seven algorithms were considered credible. To independently validate recombination signals, GARD analysis was conducted using the Beta-Gamma site-to-site variation model, specifying four rate classes under the normal run mode. This dual-method approach ensured robust detection and verification of recombination breakpoints across the WmA and WmB datasets.

### 2.3. Evolutionary Time, Contemporary Dissemination and Phylogeny

To assess temporal structure of full-length WmA (*n* = 152) and WmB (*n* = 42) genomes, root-to-tip regression analyses were performed using TempEst v1.5.3 [43]. Sequences identified as significant outliers in the regression of genetic divergence against sampling time were excluded to improve the robustness and chronological accuracy of subsequent molecular clock calibrations.

Time-scaled phylogenetic analysis, phylogeographic contemporary dissemination, and nucleotide substitution rates (substitutions per site per year) of WmCSV (WmA, WmB, and their encoded ORFs) were assessed using a Bayesian framework implemented in BEAST v10.5 [44]. In this study, two complementary approaches were employed to infer the phylogeographic spread of WmCSV. First, WmA and WmB sequences were analyzed independently after removing recombinant regions, as such regions can introduce topological inconsistencies in Bayesian phylogenetic inference. Second, to minimize sampling bias arising from unequal regional representation, the number of WmA sequences was normalized (*n* = 60) across locations to ensure a more accurate and balanced phylogenetic reconstructions—since overrepresented regions can otherwise be inferred as the most probable ancestral state at the tree root. Phylogenetic analyses were conducted using the HKY nucleotide substitution model with estimated base frequencies to account for substitution heterogeneity, a relaxed molecular clock, and the Coalescent expansion growth model as the coalescent prior. All priors, molecular clock settings, and demographic models were specified in BEAUti v10.5 to generate the BEAST XML input file for subsequent analyses. Bayesian Markov Chain Monte Carlo (MCMC) sampling was performed to estimate posterior distributions, with effective sample sizes (ESSs ≥ 200) and convergence assessed in Tracer v1.7.2 to ensure reliability of the estimates. MCMC chains were run for 1.25 × 10^9^ generations, ensuring an effective sample size (ESS) of ≥200 for all estimated parameters, thus providing robust and reliable estimates of evolutionary rates. A maximum clade credibility (MCC) tree was summarized using TreeAnnotator v10.5 and visualized in FigTree v1.4.4 (http://tree.bio.ed.ac.uk/software/figtree/, accessed 1 February 2025). To examine the temporal and spatial dynamics of virus spread, we used SpreaD (https://spreadviz.org/home, accessed 25 October 2025), which enabled visualization of the phylogeographic reconstruction and dispersal pathways.

Genomic nucleotide composition (GC/AT content) was analyzed for both components using Geneious Prime (v2023.2.1: https://www.geneious.com/), with content distribution mapped across the complete viral genomes.

### 2.4. Genetic Diversity of WmCSV Population

To investigate the genetic diversity and evolutionary dynamics of WmCSV, DNA polymorphism analysis was conducted using DnaSP v.6.12 [45], a robust tool widely used in population genetics for analyzing large genomic datasets. Nucleotide diversity (π), defined as the average number of nucleotide differences per site within a population, was calculated for all datasets, including the WmA, WmB, and all ORFs. The analysis followed established protocols [37], with π values computed using a sliding window approach (window size: 100 nucleotides; step size: 25 nucleotides) to identify regions of significant genetic variation. Significant differences in mean π values across datasets were evaluated using 95% bootstrap confidence intervals, ensuring statistical robustness in detecting polymorphic regions within the viral genomes. These analyses focused on key parameters such as the number of segregating sites (S) and the mean number of pairwise differences, providing insights into the genetic structure and evolutionary pressures acting on WmCSV populations. All analyses were performed with default settings unless otherwise specified, and results were cross validated to ensure consistency across datasets.

To assess the genetic differentiation between WmCSV global populations, pairwise *FST* values were calculated using DnaSP. *FST* values were also interpreted as an indicator of gene flow, whereby values below 0.33 were considered to reflect frequent gene flow and values above 0.33 indicated restricted gene flow, in accordance with established population genetic thresholds [46].

### 2.5. Natural Selection Inference

Selective pressures on WmCSV ORFs were assessed by calculating the non-synonymous to synonymous substitution ratio (dN/dS). Initial dN/dS estimations were performed in MEGA11 using the Nei-Gojobori method (*p*-distance). Site-specific selection was analyzed using the Datamonkey webserver (http://www.datamonkey.org/; accessed on 22 April 2025) [47] with the Fast, Unconstrained Bayesian AppRoximation (FUBAR) and Fixed Effects Likelihood (FEL) methods.

Population-level selection pressure was evaluated using DnaSP, calculating population genetic statistics such as Tajima’s D (TD) and Fu and Li’s D (FLD) tests. Mismatch distribution analysis was conducted in the DnaSP to assess the historical dynamics of the populations.

## 3. Results

### 3.1. Geographical Distribution of ToLCNDV

The typical genome organization of WmCSV is shown in Figure 1A. The global distribution using 152 WmA and 42 WmB sequences is mapped, revealing distinct geographic patterns (Figure 1B). WmCSV was identified in the Middle East (Lebanon, Israel, Palestine, Jordan, Iran, Saudi Arabia, Oman), North Africa (Sudan), and North America (USA, Mexico). In the Middle East, Jordan reported the highest number of WmA (57 sequences) and just 3 sequences of WmB. Israel harbored second the greatest number of WmA (51), while with just a single WmB reported sequence. While Iran and Oman showed a balanced distribution, with WmCSV-A and WmCSV-B each comprising 50% of the viral population. Saudi Arabia followed a similar trend with WmCSV-A at 55% and WmCSV-B at 45%. In North America, Mexico reported the 5 WmA and 6 WmB sequences and the USA reported 2 WmA and 5 WmB sequences (Table 1).

### 3.2. Evolutionary Time Estimation

Root-to-tip regression analyses were conducted to evaluate the temporal signal and evolutionary rates of WmCSV isolates, WmA and WmB, by plotting genetic divergence (substitutions per site) against sampling year (Figure 2). For WmA, the analysis revealed a substitution rate of 4.34 × 10^−4^ substitutions/site/year, with an estimated time to the most recent common ancestor (tMRCA) of approximately 1981 (x-intercept = 1980.64) and a moderate fit to the linear model (R^2^ = 0.32). In comparison, WmB exhibited a substantially higher evolutionary rate of 6.43 × 10^−3^ substitutions/site/year, an earlier tMRCA around 1962 (x-intercept = 1961.63), and a slightly stronger temporal signal (R^2^ = 0.35) (Figure 2B). These results indicate that WmB displays higher evolutionary divergence than WmA, as evidenced by its faster substitution rate, suggesting accelerated genetic evolution and potentially greater adaptability or variability in response to environmental pressures within this group.

### 3.3. Phylogeny and Phylogeography of WmCSV

#### 3.3.1. Phylogeny of WmCSV

Bayesian time-scaled phylogeny of WmCSV isolates (Figure 3) revealed geographically and temporally structured clades. The MCC tree, summarized from 10% burn-in of 1 × 10^8^ MCMC generations (ESS > 200 for all parameters), revealed a root posterior probability of ~1.0 (95% highest posterior density [HPD]). The WmA phylogeny revealed well-supported, geographically structured clades, with consistently higher posterior probabilities except for one node showing lower posterior support (value = 35). This isolated low value likely reflects limited phylogenetic signal or sequence variability within that lineage. Overall, the tree underscores clear patterns of local diversification alongside evidence of global dispersal. Within the Middle East, Saudi Arabian, Palestinian, Jordanian, Omani, and Israeli isolates formed distinct clusters, reflecting independent diversification. Some of Palestinian and Israeli isolates formed a common clade, indicating subsequent regional spread. Notably, Mexican and US isolates formed multiple subclades, reflecting ongoing viral evolution, local adaptation, and recent common ancestry and possible viral exchange between these regions. Additionally, the results of a separate analysis of WmA (*n* = 60) sequences aligned with full dataset results, indicating that the spatial signal was strong enough to overcome sampling heterogeneity.

WmB phylogeny exhibited a broader distribution and formed two major clades (Figure 3). The upper clade comprised 4 subclades with isolates from Saudi Arabia, Mexico, USA, Jordan, Palestine, Sudan, and Israel. Two of these subclades have sequence from Mexico, Jordan, and Israel, indicating common ancestry and possible viral exchange between these regions. Nonetheless, two Saudi Arabian isolates formed a separate clade. Likewise, the lower clade comprised 4 subclades, comprised WmB isolates from Saudi Arabia, Oman, and Iran clustered in a separate, highly supported lineage, indicating strong regional adaptation and divergence. All the Saudi Arabian isolates formed a separate clade, suggesting strong regional adaptation and conservation.

#### 3.3.2. Phylogeographic Spread of WmCSV

Both the methods of phylogeographic reconstruction, the one with all the sequences (*n* = 152) and the other with selected sequences (*n* = 60), suggested that WmA originated in the USA and entered Western Saudi Arabia (Al-Lith, Jeddah) around 1870s. The results of all the sequences (Figure 4) and the selected sequences (*n* = 60) are presented (Appendix A). It spread south to Jazan and into Palestine after the early 1900s, driven by regional trade and agricultural intensification. Nonetheless, WmA exhibited Saudi Arabia as an epicenter. From this epicenter, the virus dispersed northward to Palestine, Jordan, Israel (1900s) and eastward to Saudi Arabia (1960s), followed by further expansion to Iran (1980s). A westward trajectory extended to Sudan by 1980, reflecting regional trade and agricultural exchange. Transcontinental movement to Mexico occurred via Jordan, with a notable introduction around 2010, followed by a secondary spread to the USA by 2020. These pathways suggest a gradual eastward and westward expansion from the Middle Eastern core, with recent global dissemination likely facilitated by infected plant material or vector (whitefly) movement.

WmB shared a distinct origin (Figure 4), tracing back to Eastern Saudi Arabia (AlAhsa region) in the early 1800s. From there, it likely spread to Westward, Jeddah, around 1820, marking its initial local expansion. By the 1830s, a long trajectory extended to the USA during the 1830s, where both viral components (WmA and WmB) likely had an opportunity for component capture or reassortment. Subsequently, WmB continued to spread regionally, reaching Mexico by 1845. Within the Middle East, dispersal followed a similar pattern to WmA, with an inferred early introduction to Sudan by 1880, and then to Palestine/Jordan (1920), followed by Iran around 1930 and further expansion to Oman (1975). A notable reverse trajectory extended to USA from Mexico during the early 21st centuries, indicating ongoing bidirectional movement and re-emergence across continents.

### 3.4. Genetic Diversity Indices of WmCSV

#### 3.4.1. GDIs in Country-Wise WmCSV Populations

Genetic diversity metrics for WmCSV populations were assessed across geographic datasets for the two viral components, WmA and WmB, revealing distinct patterns of nucleotide variability and neutrality (Table 2).

The global WmA dataset (WmA-all, *n* = 152) revealed moderate levels of diversity (π = 0.0119; Hd = 0.996) with 21 InDels and 660 segregating sites. The haplotype number was high (h = 125), and negative values of TD (−2.22) and FLD (−6.88) suggested an excess of low-frequency polymorphisms, consistent with the recent population expansion following a selective sweep or a rapid population growth from an initial low frequency. Regionally, the WmA populations showed considerable variation. The Saudi Arabian population displayed the highest nucleotide diversity (π = 0.0205), while populations in Mexico (0.0058), Israel (0.0059), Oman (0.0068), and Jordan (0.0068), showed much lower diversity. Notably, the populations in Israel and Jordan showed significant negative (TD = −2.33* (*p* ≤ 0.05, −2.11** (*p* ≤ 0.001), and FLD = −4.33* (*p* ≤ 0.05), −3.17** (*p* ≤ 0.001), respectively) values for neutrality statistic, consistent with the global pattern of recent expansion. In contrast, the populations from Saudi Arabia, Mexico, Oman, and Palestine showed non-significant, neutral values for these tests, indicating they may be in mutation-drift equilibrium or have a more stable demographic history.

In contrast, the global WmB dataset (WmB-all, *n* = 42), displayed markedly higher genetic diversity, including 63 InDels, S = 511, and a π of 0.0508—over fourfold that of WmA-all—alongside near-maximal Hd (0.999) and k (137.45) across 40 haplotypes. Neutrality tests showed less pronounced deviations (TD = –1.52; FLD = –2.19), potentially reflecting balanced selection or stable population sizes. Regional populations consistently showed high haplotype diversity (Hd = 0.90–1.00), with Saudi Arabia (π = 0.0504; h = 9) and the USA (π = 0.0435; h = 4) recording the highest nucleotide diversities. Mexico (π = 0.0304) and Iran (π = 0.0265) also exhibited elevated variation. None of the regional populations deviated significantly from neutrality based on TD or FLD, suggesting that the signature of expansion is most clearly detected at the global, rather than regional, level for WmB.

Comparatively, WmB populations exhibited substantially greater genetic diversity and evolutionary divergence than WmA populations, as reflected by higher nucleotide diversity, greater numbers of InDels and segregating sites, and consistently elevated haplotype richness. These findings highlight contrasting evolutionary dynamics between the two lineages, with WmB populations appearing more diverse and heterogeneous, while WmA shows signals of selective constraints and demographic expansion.

#### 3.4.2. GDIs in WmCSV-Encoded ORFs

Analysis of nucleotide diversity across all positions within the WmCSV-encoded ORFs revealed distinct patterns of variation (Table 3). Among individual ORFs, the Rep gene (AC1) showed the highest diversity (π = 0.0131, Hd = 0.988), followed closely by TrAP (AC2) with π = 0.0142, reflecting elevated variability in replication and transcriptional activation functions. In contrast, the AC4 gene exhibited the lowest diversity (π = 0.0037, Hd = 0.423), consistent with strong functional constraints. Coat protein (AV1) maintained moderate diversity (π = 0.0104, Hd = 0.977), while AV2 displayed relatively low levels of variation (π = 0.0069, Hd = 0.807). These results suggest differential evolutionary pressures across WmA ORFs, with replication-associated genes displaying higher diversity.

Notably, the BC1 (MP) and BV1 (NSP) genes exhibited exceptionally high nucleotide diversity (π = 0.0367 and 0.058, Hd = 0.987 and 0.977, respectively), far surpassing any ORF in the WmA. Overall, these findings highlight a stark contrast in evolutionary dynamics: the WmB genome and its ORFs are undergoing rapid diversification.

#### 3.4.3. Nucleotide Diversity Across the Whole Genome

Nucleotide diversity (π) plots across all the genome positions for WmA and WmB populations across various countries, alongside combined datasets (WmA-All and WmB-All), unveiled distinct patterns of genetic variation (Figure 5). WmA displayed moderate genetic diversity (π = 0.00–0.10), with WmA-All and Saudi Arabian isolates displaying higher diversity in AV2, CP (AV1), TrAP (AC2), and AC4 regions, suggesting evolutionary pressure on these functional segments. Saudi Arabia exhibited the highest π peaks, especially in CP and Rep (AC1), while Israel, Jordan, and Oman showed lower, more conserved π values, possibly due to localized outbreaks or limited gene flow.

Saudi Arabian WmA isolates showed elevated diversity in the AV2, CP (AV1), TrAP (AC2), and AC4 regions, suggesting intensified evolutionary pressure on these functionally critical genomic segments. Country-wise, π values varied markedly, with Saudi Arabia exhibiting the highest peaks, particularly in the CP and Rep (AC1) coding regions, indicative of significant genetic variability. Conversely, WmA populations from Israel, Jordan, and Oman showed consistently lower π values across most genomic regions, reflecting greater sequence conservation, possibly due to localized outbreaks or restricted gene flow.

By comparison, WmB exhibited higher diversity (up to 0.14), with consistent patterns across Jordan, Mexico, the USA, Saudi Arabia, and Oman. Diversity peaks were pronounced in non-coding regions, the N-terminal of NSP (BV1), and the C-terminal of MP (BC1). These findings indicate stronger diversification in WmB, likely reflecting a deeper evolutionary history and gene flow, whereas WmA appears shaped by recent bottlenecks or selective sweeps. Additionally, these differential evolutionary dynamics between WmCSV’s genomic components highlight the need for targeted surveillance in high-diversity regions like Oman and Saudi Arabia to manage its impact on cucurbit crops.

#### 3.4.4. Genetic Differentiation Between WmCSV Populations

Pairwise *Fst* values for geographic groups of WmA and WmB isolates are shown in Figure 6. In WmA, clear genetic structuring emerged, with the sharpest divergence between Oman and Mexico (0.72), followed by Oman–Jordan (0.69) and Oman–Palestine/Israel (0.68 each). Iranian isolates also stood out, showing elevated differentiation from Mexico, Palestine, Jordan, and Israel. Yet, 12 of 28 comparisons fell below 0.33, hinting at moderate gene flow that occurs between these populations.

In contrast, WmB populations exhibited generally lower genetic differentiation. Several comparisons approached zero—or even dipped negative (e.g., USA–Mexico, –0.09; Jordan–USA, 0.01)—indicating negligible subdivision and strong connectivity. Still, Oman again stood apart, with higher divergence against Mexico (0.56), Europe (0.50), and the USA (0.40). Here, 13 of 28 values were below 0.33, underscoring greater homogeneity compared to WmA, though with a small degree of gene flow.

Overall, these results pinpointed that WmCSV population in Oman harbored higher degree of gene flow and differentiation than other regions.

### 3.5. Recombination Analysis

To unravel the recombination landscape of WmCSV genomes, initially RDP was employed to infer potential recombination breakpoints, followed by validation using the GARD analysis, ensuring a robust and precise identification of recombination events driving viral evolution.

#### 3.5.1. RDP

RDP analysis of the global WmA dataset (*n* = 152) identified a single statistically supported recombination event, indicating this mechanism is infrequent for this virus. The event was localized to nucleotide positions 2650–2760 in a Jordanian isolate (KM820195). Intriguingly, the evidence suggests this was an intraspecific recombination with another WmCSV isolate (OR865131) that was likely a recombinant with an as-yet-unidentified begomovirus (Table 4).

In contrast to WmA, 27 WmB isolates exhibited 34 potential recombination breakpoints, primarily localized within hotspot nucleotide regions: 1–200, 900–1000, 1600–1800, 2100–2200, and 2500–2700 (Table 4). The highest number of breakpoints (12) was observed in Omani isolates, with four isolates (HG962288, MH329673, MH329674, and MH329675) each showing two recombination events. In Saudi Arabia, five WmB isolates displayed six recombination events, with isolate KU360594 harboring two breakpoints. Similarly, five Iranian isolates exhibited six breakpoints, with KT272772 showing two events and the others each having one. Five Mexican isolates (PP622788–PP622791, KY124281) collectively showed five breakpoints, each with one event in the 2100–2170 region. Additionally, single isolates from Sudan (AJ245651), Jordan (JX131284), and Israel (EF201810) each presented one breakpoint. Both intra- and interspecies recombination events were detected, involving other begomovirus species, highlighting the extensive recombination dynamics in WmB and its implications for viral evolution and adaptation.

#### 3.5.2. GARD

Recombination analysis using GARD was largely consistent with RDP results. In WmA, 2730 potential breakpoints were detected across 3626 models, with just three deemed credible based on substantial improvements in corrected Akaike Information Criterion (∆C-AIC) compared to both the null model (∆C-AIC = 9828.22) and the single-tree multiple-partition model (∆C-AIC = 9066.98). These credible TA breakpoints clustered around the nucleotide positions of 1650 and 2650 (Figure 7).

Similarly, analysis of WmB sequences revealed 843 potential breakpoints from 56,143 models, with five deemed credible (∆C-AIC = 1428.40 compared to the null model and ∆C-AIC = 1221.47 compared to the single-tree multiple-partition model). These credible WmB breakpoints were primarily located around nucleotide positions 50–150, 300–400, 600–700, 1900–2100, 2300–2400, and 2700 (Figure 7).

Country-specific GARD analysis (Appendix A) revealed frequent recombination breakpoints in Saudi Arabian, Iranian, and Omani isolates. In WmA, breakpoints were distributed across AV2, CP, Rep, REn, and AC4. Saudi and Omani isolates showed multiple clusters between 400–700 bp and 1700–2200 bp (CP and Rep), while Iranian isolates had sparse breakpoints near 100–200 and 900–1000 bp (AV2, CP). Israeli isolates carried three breakpoints in REn and Rep, and Palestinian isolates had one in AV2 (150 bp), two in CP (380–420 bp), and one in REn (1000 bp). Recombination events increased after 2000, particularly in Saudi Arabia and Oman.

In WmB, recombination was more frequent, with hotspots at 200–600 bp (NSP), 200–2100 bp (MP), and 2560 bp. USA, Iranian, and Saudi isolates showed the highest densities at 600, 2000, and 2600 bp, while Jordanian and Mexican isolates clustered at 500–600, 800, and 2000–2300 bp (Appendix A). Unlike WmA, WmB recombination events were temporarily evenly distributed.

### 3.6. Nucleotide Substitution Rate

Mean nucleotide substitution rates for WmA and WmB were estimated using both strict and relaxed uncorrelated molecular clocks, with 95% HPD intervals. Substitution rates differed between the components: WmA exhibited rates of 1.44 × 10^−3^ (strict clock) and 1.22 × 10^−3^ (relaxed clock), while WmB showed higher rates of 1.26 × 10^−4^ (strict clock) and 1.23 × 10^−4^ (relaxed clock).

### 3.7. Selection Pressure Inference Analysis

#### 3.7.1. Neutrality Indices

Selection pressure analyses of WmCSV ORFs using FEL and FUBAR (Table 5) showed that most fitted the Kimura 2-parameter + gamma (K2 + G) model, except REn (T92 + G) and AC4 (JC). Mean genetic distances ranged from 0.0035 ± 0.0057 (AV2) to 0.0147 ± 0.0025 (TrAP), indicating variable divergence across ORFs.

Overall, dN/dS ratios were >1 (except AV2 and AC4), suggesting a strong and relaxed purifying selection as the dominant force. In WmA, CP had the highest dN/dS (2.72), Rep showed moderate constraints (1.35), and TrAP and REn evolved near-neutral (1.28 and 1.21). In WmB, divergence was greater: NSP had the highest synonymous rate (dS = 0.15 ± 0.028), while MP showed a dN/dS of 2.55, consistent with potential positive selection.

Neutrality tests further distinguished the two components. WmA ORFs, especially Rep, showed significant purifying selection (TD = −2.46; FLD = −5.69), whereas WmB yielded non-significant results, reflecting weaker selective pressures. Site-specific analyses revealed widespread negative selection in WmA (e.g., Rep: 39 FEL, 14 FUBAR sites). Positive selection was rare, detected only in the TrAP gene (with two sites identified by FUBAR) and in one site each in AV2, CP, and REn (identified by FEL). In WmB, MP harbored 69 negatively selected and 1 positively selected sites, consistent with near-neutral evolution.

#### 3.7.2. Mismatch Distribution

Mismatch distribution analysis (Appendix A) was used to infer demographic changes in WmCSV populations and their ORFs. For WmA (Appendix A), the overall distribution was ragged and multimodal, deviating from the unimodal curve expected under recent expansion. Most ORFs showed bimodal or multimodal patterns, consistent with stable or declining population sizes. Exceptions were AV2 (Appendix A), which displayed a unimodal curve indicative of recent expansion, and AC4 (Appendix A), which was skewed toward very low pairwise differences, reflecting limited variation.

In contrast, WmB (Appendix A) showed an irregular, multimodal distribution with a broad range of pairwise differences, suggesting a more complex demographic history. Both NSP and MP ORFs (Appendix A) exhibited multimodal distributions with multiple peaks, consistent with dynamic evolution involving recombination or multiple introductions.

## 4. Discussion

The present study provides a comprehensive genomic, evolutionary, and phylogeographic assessment of WmCSV, with particular focus on its bipartite genome components, WmA and WmB. The findings demonstrated significant heterogeneity in global distribution, evolutionary dynamics, genetic diversity, and recombination patterns, underscoring the complex biology of this virus and its capacity to adapt to diverse agro-ecological landscapes.

Geographical analysis confirmed WmCSV’s establishment in the Middle East and more recent introduction to North America (USA and Mexico), consistent with earlier reports of its emergence in Yemen and Sudan [8,9]. The imbalance in available sequences—152 WmA versus 42 WmB, especially in Jordan and Israel—may reflect a sampling bias where research and surveillance efforts have historically prioritized the sequencing of the DNA-A component, which contains the core replication genes, over the DNA-B [48,49]. By contrast, more balanced WmA/WmB ratios in Iran, Oman, Saudi Arabia, and the Americas suggest either a richer pool of WmB variants or region-specific evolutionary dynamics. While uneven sampling can theoretically bias root inference, our model robustly identified distinct origins for WmA (United States) and WmB (Saudi Arabia) despite most sequences coming from Palestine/Israel. Additionally, analysis of a smaller, regionally balanced dataset of 60 WmA sequences aligned with full dataset results. This indicates that the spatial signal was strong enough to overcome sampling heterogeneity, reinforcing confidence in the inferred origins and migration pathways.

A central and striking finding of this study is the profound evolutionary asymmetry between the WmA and WmB components. The WmB component exhibits a substantially higher evolutionary rate (~1.26 × 10^−4^ subs/site/year) compared to WmA (~1.44 × 10^−3^ subs/site/year), an earlier time to the most recent common ancestor (tMRCA ~1962 vs. ~1981), and over fourfold higher nucleotide diversity (π = 0.0508 vs. 0.0119). This suggests that WmB represents an older, more divergent lineage undergoing accelerated evolution, likely linked to its role in movement and symptom expression across diverse hosts [50]. Additionally, this pattern is consistent with observations in other bipartite begomoviruses, where the DNA-B component, responsible for movement and symptom determination, often evolves faster and is more genetically diverse than the DNA-A component [32,34,49]. The proteins encoded by WmB—the nuclear shuttle protein (NSP/BV1) and movement protein (MP/BC1)—interact directly with a wide array of host factors to facilitate cell-to-cell movement and suppress host defenses [7]. This constant “arms race” with the host is a potent driver of rapid evolution, leading to the elevated dN/dS ratios and high diversity we observed in these genes [27,49,51].

This accelerated evolution is further fueled by a vastly more complex recombination landscape in WmB. We detected 34 recombination events in WmB compared to a single event in WmA. Recombination hotspots were concentrated in the NSP and MP genes, which are known to be recombination-prone regions in begomoviruses [52,53]. The high frequency of recombination in WmB, particularly involving interspecies events, highlights its role as a genetic module that can be readily swapped or acquired, facilitating rapid host adaptation and virulence shifts [4,27,36,38,54]. This genetic promiscuity stands in stark contrast to the WmA component, which appears to be evolutionarily more conserved, likely due to strong functional constraints on its replication-associated proteins (Rep, REn) which must maintain precise interactions with host replication machinery. *Fst*-based genetic differentiation analyses also highlighted these contrasts. WmA displayed higher *Fst* values, indicating stronger geographic structuring and limited gene flow, with Oman particularly isolated—possibly due to co-evolution with local whitefly biotypes or host plants. Conversely, WmB showed lower or even negative *Fst* values (e.g., USA–Mexico), indicating high connectivity and long-distance movement, consistent with human-mediated spread via infected plant material or vector migration [55,56].

The population genetic analyses reinforce this narrative of divergent evolutionary paths. The significantly negative TD and FLD values for global and regional (Israel, Jordan TD = −2.33* and −2.11**), WmA populations are strong signals of a recent population expansion, potentially following a selective sweep or a bottleneck event [49,57,58]. In contrast, the WmB populations showed neutrality indices closer to zero, suggesting a more stable demographic history over a longer period or a population under balancing selection, which is often associated with genes involved in host–pathogen interactions. Selection analyses showed pervasive purifying selection across most ORFs, underscoring functional constraints critical for viral survival. However, elevated dN/dS ratios (>1) in CP, Rep, MP, and NSP suggest localized domains may be under positive selection to evade host defenses or optimize function, consistent with plant–virus coevolution [34]. ORF-level comparisons reinforced the component-specific differences: WmA’s Rep (AC1) and TrAP (AC2) exhibited moderate diversity and near-neutral evolution, while AC4 remained highly conserved due to its role in pathogenicity suppression [59,60,61]. In contrast, NSP and MP genes in WmB displayed exceptionally high variability, consistent with relaxed purifying selection and episodic positive selection. Neutrality tests confirmed these trends: WmA populations showed strong purifying selection, whereas WmB evolved under weaker constraints, supporting broader adaptability.

The phylogeographic analyses suggest intriguingly distinct origins for the two components. The reconstruction points to a potential American origin for WmA around the 1870s, with subsequent spread to the Middle East, while WmB appears to have originated in Eastern Saudi Arabia (AlAhsa region) at an even earlier date (1807). While seemingly contradictory, this could be explained by the ancient reassortment of the two components [50]. It is plausible that the WmA lineage now prevalent globally originated from a New World begomovirus that recombined with or acquired an Old World DNA-B component in an intermediate host, a phenomenon documented in the emergence of other begomoviruses [62,63]. Movement from Palestine into Israel and Jordan followed, consistent with cucurbit cultivation and whitefly outbreaks in the Jordan Valley [11,12]. The Middle East subsequently acted as a central hub for the diversification and global spread of the reassorted virus, a pattern driven by historical trade routes, modern agricultural exchange, and the expansion of its whitefly vector, *B. tabaci*, or cross-border trade [22,64]. Notably, phylogeographic evidence indicates a recent “backflow” from Palestine/Israel to the USA and subsequently Mexico, likely driven by globalization of agricultural trade and seed exchange [19].

## 5. Conclusions

In conclusion, our results depict a model of co-evolving but distinct evolutionary trajectories for the two components of WmCSV. WmA appears to be a stable, foundational genome that has recently expanded rapidly. The WmB component, by contrast, is a dynamic and highly variable module, evolving rapidly through high mutation rates and frequent recombination, likely enabling host adaptation and facilitating spread. The identification of specific geographic hotspots for diversity (Saudi Arabia) and recombination (Oman) is critical for directing future surveillance efforts. Managing the threat of WmCSV will require continued monitoring of these evolutionary dynamics, particularly the movement and reassortment of the more promiscuous WmB component, to mitigate the risk of new and more aggressive viral strains emerging.

## Figures and Tables

**Figure 1 viruses-17-01571-f001:**
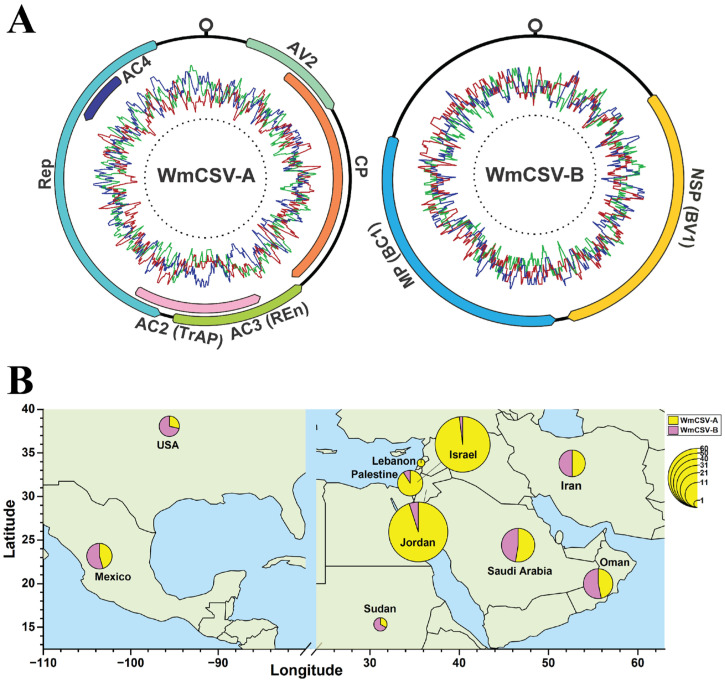
Genome organization of WmCSV (**A**). The outermost circle illustrates the positions and transcriptional orientations (indicated by arrows) of all open reading frames (ORFs) encoded on the WmA and WmB components. The origin of replication—marked by a hairpin loop structure—is also indicated. The middle circle displays the GC (blue) and AT (red and green) content distribution across the genome. Geographical distribution and population density of WmCSV in the World (**B**).

**Figure 2 viruses-17-01571-f002:**
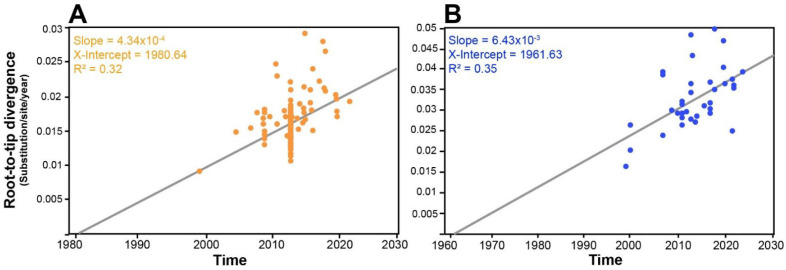
Root-to-tip regression (TempEST) analysis of WmA (**A**) and WmB (**B**) sequences. *X*-axis shows time (year), and *y*-axis shows root-to-tip divergence (substitutions/site/year). The slope of the linear regression line represents the estimated evolutionary rate, the x-intercept indicates the inferred time of the most recent common ancestor (tMRCA), and the R^2^ value quantifies the strength of the temporal signal.

**Figure 3 viruses-17-01571-f003:**
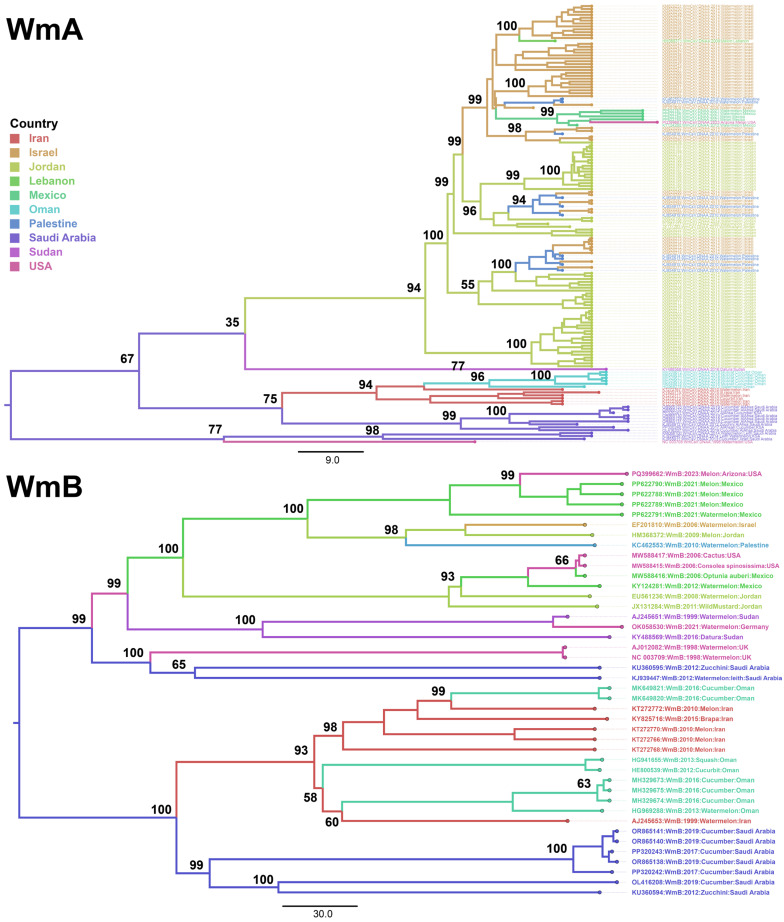
Time-scaled phylogenetic tree of complete genome sequences of WmCSV, WmA and WmB, based on Bayesian Markov Chain Monte Carlo method in BEAST (v10.5) employing a relaxed molecular clock model and a coalescent exponential growth demographic model. The HPD = 95% is mentioned in the figure. The trees were visualized using FigTree and edited employing Adobe Illustrator. Branches and tip labels are color-coded by country. Scale bars represent the number of nucleotide substitutions per site.

**Figure 4 viruses-17-01571-f004:**
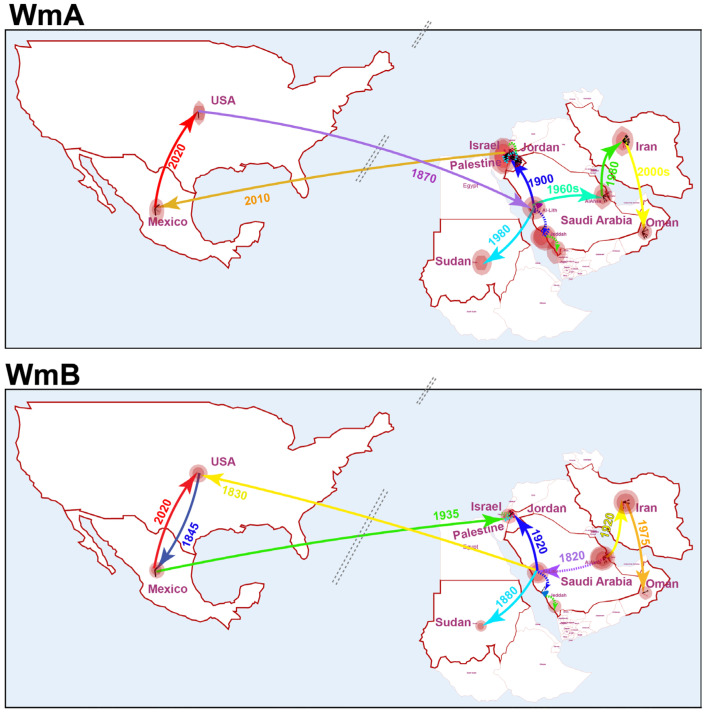
Spatiotemporal reconstruction of WmA and WmB dissemination. Phylogeographic analysis of WmA and WmB complete genomes, inferred using a Bayesian model. The map illustrates the putative geographic origins and subsequent global spread. Colored lines represent international transmission pathways, with violet and red indicating the earliest and most recent events, respectively. The size of red circles corresponds to the relative effective viral population size at each location. Local transmission events within countries are shown by dotted arrows and thin black lines. The diagonal dotted lines represent the adjustment in the location. The final layout adjustments were made in Adobe Illustrator.

**Figure 5 viruses-17-01571-f005:**
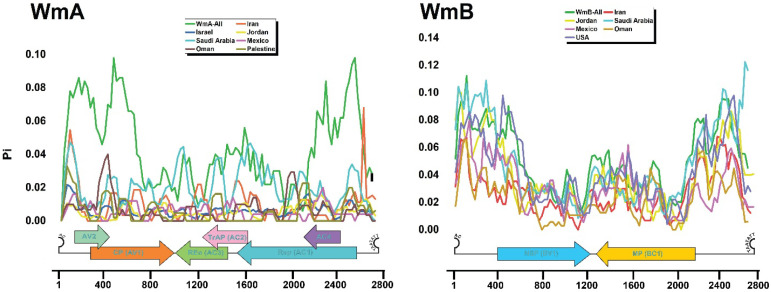
Nucleotide diversity of WmCSV DNA-A and DNA-B components. The figure illustrates the nucleotide diversity (π) across the complete genome sequences of WmA and WmB for their populations in different countries. The *x*-axis represents the nucleotide and open reading frames (ORFs) position along the viral genome. The *y*-axis represents the nucleotide diversity (π) at each position, calculated using a sliding window approach.

**Figure 6 viruses-17-01571-f006:**
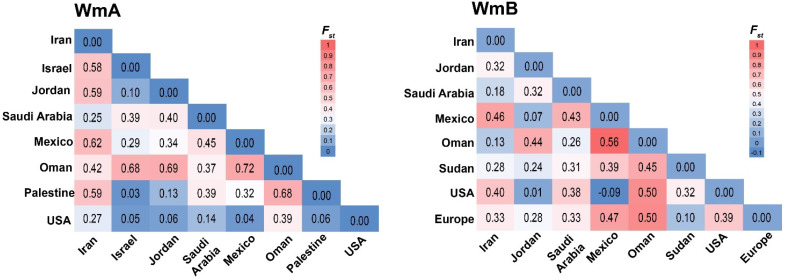
Heatmap showing pairwise *Fst* between WmCSV, both WmA (**left** panel) and WmB (**right** panel), populations across different countries.

**Figure 7 viruses-17-01571-f007:**
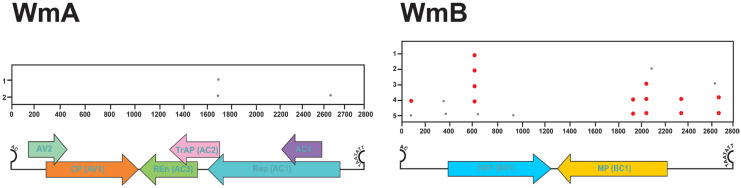
Distribution of recombination breakpoints in global datasets of WmA and WmB. High-confidence breakpoints (supported by high AICc scores) are marked by large red dots, while less reliable breakpoints (lower AICc scores) appear as small gray dots. The *x*-axis displays nucleotide positions, and the *y*-axis shows the breakpoint count. The linear genome structures of WmA and WmB are illustrated for reference.

**Table 1 viruses-17-01571-t001:** Total number of WmA and WmB sequences from various countries included in the nucleotide diversity analysis.

Country	No. of Sequences	
	WmA	WmB
Saudi Arabia	12	9
Oman	7	8
Sudan	1	2
Iran	6	6
Palestine	10	1
Lebanon	1	0
Jordan	57	3
Israel	51	1
USA	2	5
Mexico	5	6
**Total**	**152**	**42**

**Table 2 viruses-17-01571-t002:** Geographic landscape of genetic diversity in WmCSV populations

Dataset	Virus Comp	No. of Eq	InDels	S	Eta (h)	Eta (s)	Pi	ɵ−Eta	Hd	k	h	TD	FLD
WmA-all	**WmA**	152	21	660	766	426	0.0119	0.05004	0.996	144.96	125	−2.22	−6.88
Iran		6	0	144	144	124	0.019	0.0229	1.00	52.33	6	−1.10 ^ns^	−1.12 ^ns^
Israel		51	3	226	236	158	0.0068	0.0195	0.989	18.651	43	−2.33 *	−4.33 *
Jordan		57	6	175	186	97	0.0059	0.0147	0.996	16.354	53	−2.11 **	−3.17 *
Saudi Arabia		12	1	200	209	110	0.0205	0.0259	0.939	62.02	9	−0.63 ^ns^	−0.61 ^ns^
Mexico		5	2	38	38	35	0.0058	0.0063	1.00	15.80	5	−0.88 ^ns^	−0.76 ^ns^
Oman		6	0	51	51	43	0.0068	0.0081	0.933	18.67	5	−1.05 ^ns^	−1.05 ^ns^
Palestine		10	2	50	53	26	0.0062	0.0068	0.978	16.98	9	0.83 ^ns^	−0.41 ^ns^
WmB-all	**WmB**	42	63	511	689	416	0.0508	0.0854	0.999	137.45	40	−1.52	−2.19
Iran		6	3	181	192	148	0.0265	0.0308	1.00	72.13	6	−1.23 ^ns^	−0.83 ^ns^
Jordan		3	6	161	163	121	0.0392	0.176	1.00	108.0	3	N/A	N/A
Saudi Arabia		9	37	408	451	287	0.0504	0.0609	1.00	137.19	9	−0.90 ^ns^	−0.90 ^ns^
Mexico		6	6	180	188	95	0.0304	0.030	1.00	83.87	6	−0.93 ^ns^	−0.68 ^ns^
Oman		8	5	145	149	59	0.0241	0.0223	1.00	63.57	7	0.27 ^ns^	0.29 ^ns^
USA		5	10	226	232	92	0.0435	0.041	0.90	119.60	4	0.56 ^ns^	0.65 ^ns^

Abbreviations used in the table are Watermelon chlorotic stunt virus DNA-A (WmA), DNA-B (WmB), insertions and deletions (InDels), total number of polymorphic (segregating) sites (S), total number of mutations [Eta (h)], total number of singleton mutations [Eta (S)], haplotype diversity (Hd), average number of nucleotide differences between sequences (k), nucleotide diversity (π), haplotypes (h), Watterson’s estimate of the population mutation rate based on the total number of segregating sites (Ѳw), Watterson’s estimate of the population mutation rate based on the total number of mutations (Ѳ-Eta), Tajima’s D (TD), Fu and Li’s D (FLD), and non-significant (ns). * ≤ 0.05; ** ≤ 0.001.

**Table 3 viruses-17-01571-t003:** Landscape of genetic diversity in WmCSV-encoded Open Reading Frames (ORFs)

Country	Virus Comp	No. of Seq	InDel Sites	S	Eta (h)	Eta (s)	Pi (π)	ɵ from Eta	Hd	k	h
WmCSV-A	**WmA**	152	21	660	766	426	0.0119	0.05004	0.996	144.96	125
AV1 (CP)		152	3	186	207	114	0.0104	0.048	0.977	8.06	84
AV2		152	0	77	86	51	0.0069	0.043	0.807	2.47	50
AC1 (Rep)		152	3	267	321	171	0.0131	0.053	0.988	14.15	107
AC2 (TrAP)		152	0	102	123	72	0.0142	0.054	0.90	5.81	67
AC3 (REn)		152	0	94	106	62	0.0107	0.047	0.794	4.34	57
AC4		152	0	18	19	13	0.0037	0.024	0.423	0.539	18
WmCSV-B	**WmB**	42	63	811	989	445.85	0.0508	0.0854	0.999	137.45	40
BV1 (NSP		42	9	62	76	30	0.058	0.0819	0.977	12.51	27
BC1 (MP)		42	0	196	222	96	0.0367	0.0563	0.987	33.73	32

Abbreviations used in the Watermelon chlorotic stunt virus DNA-A (WmA), DNA-B (WmB), insertions and deletions (InDels), total number of polymorphic (segregating) sites (S), total number of mutations [Eta (h)], total number of singleton mutations [Eta (S)], haplotype diversity (Hd), average number of nucleotide differences between sequences (k), nucleotide diversity (π), haplotypes (h), Watterson’s estimate of the population mutation rate based on the total number of segregating sites (Ѳw), Watterson’s estimate of the population mutation rate based on the total number of mutations (Ѳ-Eta), and not calculated (n/c).

**Table 4 viruses-17-01571-t004:** Recombination events detected by RDP in WmA and WmB of WmCSV

Virus	Accession #	Recomb. Event	Breakpoints		Parents		Detection Methods ^a^	*p*-Value ^b^
			Begin	End	Major	Minor		
**WmA**	**KM820195**	**1**	**2655**	2751	Unknown	OR865131	GBCMR**S**3	1.89 × 10^−37^

WmB	AJ245651	1	2707	51	OR865138	KY124281	MCS3**L**	1.15 × 10^−10^
	EF201810	1	1689	9	Unknown	KC462553	BMC**S**L	1.21 × 10^−14^
	HE800539	1	990	1984	Unknown	OR865138	MCS3**L**	1.15 × 10^−10^
	HG941655	2	2271	2691	Unknown	KT272768	RGBCS**L**	8.62 × 10^−7^
			69	1241	Unknown	OR865138	MCS3**L**	1.15 × 10^−10^
	JX131284	1	802	1926	MW588415	HM368372	RBMCS3**L**	7.24 × 10^−8^
	KJ939447	1	2638	2721	Unknown	OR865138	GBM3**L**	6.55 × 10^−13^
	KT272766	1	990	1984	Unknown	OR865138	MCS3**L**	1.15 × 10^−10^
	KT272768	1	113	1639	Unknown	OR865138	MCS3**L**	1.15 × 10^−10^
	KT272770	1	1019	1546	KC462553	Unknown	MCS3**L**	1.15 × 10^−10^
	KT272772	2	993	1615	Unknown	OR865138	MCS3**L**	1.15 × 10^−10^
			1591	2708	OR865138	KY124281	RGBCS**L**	8.62 × 10^−7^
	HG969288	2	990	2693	OR865138	KY124281	RGBCS**L**	8.62 × 10^−7^
			990	2551	Unknown	OR865138	MCS3**L**	1.15 × 10^−10^
	KU360594	2	329	728	OR865141	AJ245653	RGBMCS**L**	1.32 × 10^−8^
			1087	2002	Unknown	PP320242	MCS3**L**	2.94 × 10^−11^
	KU360595	1	364	741	NC003709	AJ245651	RGMB3C**S**L	1.62 × 10^−10^
	KY825716	1	2483	52	OR865138	KY124281	RGBCS**L**	8.62 × 10^−7^
	MH329673	1	2483	52	OR865138	KY124281	RGBCS**L**	8.62 × 10^−7^
	MH329674	2	2694	720	KY124281	OR865138	RGBCS**L**	8.62 × 10^−7^
			721	2482	Unknown	OR865138	MCS3**L**	1.15 × 10^−10^
	MH329675	2	2483	52	OR865138	KY124281	RGBCS**L**	8.62 × 10^−7^
			990	1984	Unknown	OR865138	MCS3**L**	1.15 × 10^−10^
	MK649820	2	2581	437	OR865138	KY124281	RGBCS**L**	8.62 × 10^−7^
			990	1984	Unknown	OR865138	MCS3**L**	1.15 × 10^−10^
	MK649821	1	990	1984	Unknown	OR865138	MCS3**L**	1.15 × 10^−10^
	OR865141	1	2607	13	OR865138	NC003709	GMC3**L**	8.10 × 10^−13^
	PP320242	1	2608	2716	OR865138	Unknown	RGMCS3**L**	1.19 × 10^−36^
	PP320243	1	2609	565	OR865138	MH329674	GBCS3**L**	5.61 × 10^−35^
	PP622788	1	2203	50	PQ399662	KY124281	RGBMCS3**L**	4.33 × 10^−22^
	PP622789	1	2218	77	PQ399662	KY124281	RGBMCS3**L**	4.33 × 10^−22^
	PP622790	1	2168	70	PQ399662	MW588415	RGBMCS3**L**	4.33 × 10^−22^
	PP622791	1	486	2398	PQ399662	KY124281	MCS3**L**	2.51 × 10^−8^

^a^ B, Bootscan; C, Chimaera; G, GeneConv; L, LARD; M, MaxChi; P, Phylpro; R, RDP; S, SisScan; 3, 3SEQ. ^b^ The lowest *p*-value corresponds to the recombination program (bold) is mentioned.

**Table 5 viruses-17-01571-t005:** Demographic history analysis, Neutrality indices (TD and FLD test), of WmCSV.

Dataset	ORF	Best Model	Mean Distance (d)	dN	dS	dN/dS	TD	FLD	Expansion	Selection Sites			
										FEL		FUBAR	
										NS	PS	NS	PS
**WmA**	**WmA-all**	**K2 + G**	**--**	**--**	**--**	**--**	**−2.22 ****	**−6.88 ***	Yes	--	--	--	--
	AV2	JC + G	0.0035 ± 0.0057	0.006 ± 0.001	0.0105 ± 0.0025	0.58	−2.64 **	−6.70 *	Yes	3	0	6	1
	CP (AV1)	K2 + G	0.0106 ± 0.0018	0.0039 ± 0.0006	0.0313 ± 0.0061	2.72	−1.32 **	−5.80 *	Yes	30	0	27	1
	Rep (AC1)	K2 + G	0.0135 ± 0.0014	0.01 ± 0.0012	0.0203 ± 0.0039	1.35	−2.46 **	−5.69 *	Yes	39	1	14	1
	TrAP (AC2)	K2 + G	0.0147 ± 0.0025	0.0115 ± 0.0025	0.0257 ± 0.0064	1.28	−2.35 *	−5.99 *	Yes	8	0	8	2
	REn (AC3)	T92 + G	0.011 ± 0.002	0.0091 ± 0.0024	0.0166 ± 0.0043	1.21	−2.45 **	−5.87 *	Yes	2	1	5	2
	AC4	JC	0.0038 ± 0.0014	0.0038 ± 0.0017	0.004 ± 0.0022	1.0	−1.30 *	−2.81 *	Yes	0	0	0	0
	Iran	T92	--	--	--	--	−1.10 ^ns^	−1.12 ^ns^	No	--	--	--	--
	Israel	K2 + G	--	--	--	--	−2.33 *	−4.33 *	Yes	--	--	--	--
	Jordan	K2 + G	--	--	--	--	−2.11 *	−3.17 *	Yes	--	--	--	--
	Saudi Arabia	K2 + G	--	--	--	--	−0.63 ^ns^	−0.61 ^ns^	No	--	--	--	--
	Mexico	JC	--	--	--	--	−0.88 ^ns^	−0.76 ^ns^	No	--	--	--	--
	Oman	K2	--	--	--	--	−1.05 ^ns^	−1.05 ^ns^	No	--	--	--	--
	Palestine	JC	--	--	--	--	0.83 ^ns^	−0.41 ^ns^	No	--	--	--	--
**WmB**	WmB-all	GTR + G	--	--	--	--	−1.52 ^ns^	−2.00 ^ns^	No	--	--	--	--
	NSP (BV1)	GTR + G	0.0626 ± 0.0093	0.0338 ± 0.0081	0.15 ± 0.028	1.85	−1.07 ^ns^	−1.29 ^ns^	No	5	0	23	0
	MP (BC1)	HKY + G	0.0382 ± 0.0034	0.015 ± 0.0031	0.0114 ± 0.011	2.55	−1.30 ^ns^	−1.86 ^ns^	No	58	0	69	1
	Iran	T92 + G	--	--	--	--	−1.23 ^ns^	−0.83 ^ns^	No	--	--	--	--
	Jordan	T92	--	--	--	--	N/A	N/A	--	--	--	--	--
	Saudi Arabia	T92 + G	--	--	--	--	−0.90 ^ns^	−0.90 ^ns^	No	--	--	--	--
	Mexico	T92 + G	--	--	--	--	−0.93 ^ns^	−0.68 ^ns^	No	--	--	--	--
	Oman	T92	--	--	--	--	0.27 ^ns^	0.29 ^ns^	No	--	--	--	--
	USA	T92	--	--	--	--	0.56 ^ns^	0.65 ^ns^	No	--	--	--	--

Abbreviations used in the table are Watermelon chlorotic stunt virus DNA-A (WmA), DNA-B (WmB), coat protein (CP), replication-associated protein (Rep), transcriptional activator protein (TrAP), replication enhancer protein (REn), movement protein (MP), nuclear shuttle protein (NSP), Tajima’s D test (TD), Fu and Li’s D test (FLD), positively selected sites (PS), negatively selected sites (NS), and non-significant (ns). * ≤0.05; ** ≤0.001.

## Data Availability

This article and its Appendix A files contain all data generated or analyzed during this study, apart from the WmCSV sequences. These sequences are available upon request from the corresponding author.

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
