# Peer review of "Contrasting Evolutionary Dynamics and Global Dissemination of the DNA-A and DNA-B Components of Watermelon Chlorotic Stunt Virus"

_viruses, 2025, doi:10.3390/v17121571_

Round 1
Reviewer 1 Report
Comments and Suggestions for Authors
The manuscript titled "Contrasting Evolutionary Dynamics and Global Dissemination of the DNA-A and DNA-B Components of Watermelon chlorotic stunt virus" though comes under the scope of MDPI journal Viruses, it doesn't qualify for the publication as it doesn't add any significant findings in the field of Virology. The author(s) can consider adding additional experiments to increase the value of their report.
Author Response
Comment 1. The manuscript titled "Contrasting Evolutionary Dynamics and Global Dissemination of the DNA-A and DNA-B Components of Watermelon chlorotic stunt virus" though comes under the scope of MDPI journal Viruses, it doesn't qualify for the publication as it doesn't add any significant findings in the field of Virology. The author(s) can consider adding additional experiments to increase the value of their report.
Response 1: Thank you for your feedback. This work presents the “first comprehensive global comparison” of the DNA-A and DNA-B of WmCSV, revealing contrasting evolutionary dynamics that had not been previously documented. Our results show that DNA-B exhibits substantially higher genetic diversity, recombination frequency (34 vs. 1 events), and a distinct evolutionary origin, identifying Saudi Arabia as a potential ancestral source and the Middle East as a regional hub of diversification. These findings provide new insights into begomovirus evolution, adaptation, and global movement, advancing understanding of how bipartite genomes evolve under different selective pressures.
This study offers a novel framework for virological and epidemiological research, informing future surveillance and resistance-breeding strategies. Author believes these contributions represent a meaningful advancement in plant virology, particularly in understanding virus emergence in arid agroecosystems.
Reviewer 2 Report
Comments and Suggestions for Authors
The manuscript by Iqbal investigates the evolutionary dynamics and global dissemination of watermelon chlorotic stunt virus (WmCSV), a bipartite begomovirus that represents a significant and growing threat to global cucurbit and watermelon production. This is a highly relevant topic, as robust Bayesian phylogenetic analyses can provide critical insights into the epidemiology and evolution of this pathogen.
Major concerns:
The study's impact is currently limited by insufficient methodological detail. The Methods section requires substantial revision and expansion to ensure the analyses are transparent, reproducible, and can be critically evaluated by the scientific community. The following specific points must be addressed:
(1) Treatment of recombination in Bayesian phylogenetics: The study appears to have included recombinant sequences in its Bayesian phylogenetic analysis. Standard practice is to use recombination-free alignments for such analyses to avoid topological errors. While novel BEAST modules can model recombination (e.g., Hoscheit & Desbiez, 2025, 10.1016/j.meegid.2025.105732), their application remains challenging. The authors must justify their approach or re-analyze their data using conventional, recombination-free genomic regions.
(2) Recombination screening prior to selection analysis: To minimize false positives, sequence alignments must be rigorously screened for recombination before conducting tests of positive selection. As demonstrated previously (e.g., Sironi et al., 2015), genetic variation introduced by recombination can be misinterpreted as positive selection, confounding the results.
(3) Threshold of recombination detection: The use of default parameters in the RDP suite is a significant concern, as it is prone to false positives, especially in rapidly evolving viruses. The analysis should be repeated using more stringent criteria, where recombination events are only considered reliable if identified by a majority of the suite's algorithms (e.g., at least four out of seven) with a conservative p-value cutoff (e.g., < 1.0E-5), as established in recent literature (Gibbs et al., 2017, DOI: 10.1093/ve/vex002; BenMansour, et al, 2023, Plant Pathology, DOI: 10.1111/ppa.13824; Long, et al, 2024, DOI: 10.3390/v16121850).
(4) Accounting for heterochrony in classical population cenetics: Classical population genetic analyses assume a contemporaneous sample, which is often violated in datasets spanning long time periods (this study in this case). Even minor heterochrony can substantially distort polymorphism distributions, leading to an overestimation of diversity, an excess of rare mutations, and reduced linkage disequilibrium. The authors should address this potential bias, for instance, by employing unbiased estimators as suggested by Depaulis et al. (2009, DOI:10.1371/journal.pone.0005541.g001), to ensure their conclusions are valid.
(5) The utilized data set for phylogeographic analysis in this study is highly unbalanced. This would result in an inaccurate reconstruction of the origin of the sampled WmCSV diversity, as the phylogenetic reconstruction would tend to identify the most sampled location as the most probable state in the root of the tree. This is the case here, as well the inferred migration pathways.
Minor concern
Page 3 of 23
(1) Subsections 2.2 (L109-129) and 2.4 (L152-163) should be combined. Both sections report results (MRCA and substitution rates) derived from the same Bayesian phylogenetic analysis. Presenting them together would improve the manuscript's logical flow and avoid unnecessary fragmentation.
(2) What method was used for the selection of molecular clock model and tree prior? Ideally, this would be marginal likelihoods via path sampling/stepping-stone sampling. Please clarify.
Page 4 of 23
L166-171 There is a confusion in the paragraph between natural selection and demographic history inference. The analyses performed are for natural selection analysis, not for inferring demographic history. The methods used—mismatch distribution and neutrality tests (Tajima's D and Fu's Fs)—are designed to detect population expansion under a neutral model. However, they do not directly reconstruct historical population size changes. To explicitly infer demographic history, a dedicated approach such as the Bayesian Skyline Plot (BSP) is required.
Author Response
Comment 1. Treatment of recombination in Bayesian phylogenetics: The study appears to have included recombinant sequences in its Bayesian phylogenetic analysis. Standard practice is to use recombination-free alignments for such analyses to avoid topological errors. While novel BEAST modules can model recombination (e.g., Hoscheit & Desbiez, 2025, 10.1016/j.meegid.2025.105732), their application remains challenging. The authors must justify their approach or re-analyze their data using conventional, recombination-free genomic regions.
Response 1: Author agreed to the reviewer that recombination sequence can pose topological errors in Bayesian phylogeny. Author re-analyzed the phylogenies, both for WmA and WmB, after removing the recombinant regions from the sequences.
The initial analysis of WmA identified only one credible recombinant region. When this region was removed and the analysis was repeated, the results showed only minor differences. For example, one Saudi isolate (KM066100) grouped differently, clustering with a Palestinian isolate. Likewise, one isolates from Oman and Iran grouped together. However, since these changes were not substantial, the original findings were maintained.
For WmB, the revised analysis showed more significant changes, including alterations in the grouping of some sequences. However, the overall number of major and sub-clades remained consistent (see the updated Figure 3 in the manuscript). The relevant sections on MM (Lines 118-127), results (Lines 301-312), and the figure have been amended to reflect this.
Comment 2. Recombination screening prior to selection analysis: To minimize false positives, sequence alignments must be rigorously screened for recombination before conducting tests of positive selection. As demonstrated previously (e.g., Sironi et al., 2015), genetic variation introduced by recombination can be misinterpreted as positive selection, confounding the results.
Response 2: Following the identification of recombination signals in the AV2 region of WmA and MP and NSP of WmB, the author repeated the FEL and FUBAR analyses on these open reading frames (ORFs) after excluding the recombinant sequences. This revision slightly altered the outcome in negative sights only, as no positive selection sites had been detected in these regions initially. The updated results are shown in Table 5.
Author found this suggestion highly important and will be practicing it in futuristic studies.
Comment 3. Threshold of recombination detection: The use of default parameters in the RDP suite is a significant concern, as it is prone to false positives, especially in rapidly evolving viruses. The analysis should be repeated using more stringent criteria, where recombination events are only considered reliable if identified by a majority of the suite's algorithms (e.g., at least four out of seven) with a conservative p-value cutoff (e.g., < 1.0E-5), as established in recent literature (Gibbs et al., 2017, DOI: 10.1093/ve/vex002; BenMansour, et al, 2023, Plant Pathology, DOI: 10.1111/ppa.13824; Long, et al, 2024, DOI: 10.3390/v16121850).
Response 3: Author has already adopted the suggested strategy for the RDP analysis. Rather thans suggested four algorithms, a more conservative threshold was used: only recombination events detected by at least five algorithms with a p-value of less than 0.05 were considered credible. Please see lines 196-198.
Comment 4. Accounting for heterochrony in classical population cenetics: Classical population genetic analyses assume a contemporaneous sample, which is often violated in datasets spanning long time periods (this study in this case). Even minor heterochrony can substantially distort polymorphism distributions, leading to an overestimation of diversity, an excess of rare mutations, and reduced linkage disequilibrium. The authors should address this potential bias, for instance, by employing unbiased estimators as suggested by Depaulis et al. (2009, DOI:10.1371/journal.pone.0005541.g001), to ensure their conclusions are valid.
Response 4: Author thanks the reviewer for raising this important point. I agree that this is a key consideration for any temporal dataset. In this study, author sought to mitigate this concern by employing a multifaceted analytical strategy. Specifically, author complemented standard diversity indices with pairwise Fst analysis, a method that is relatively robust to the effects of temporal sampling when comparing population structure. The consistent patterns of genetic differentiation revealed by Fst across sampling intervals corroborate the findings from other analyses, providing a convergent and reliable signal of the population dynamics described. While I acknowledge that some diversity estimates could be influenced by the time-span of sampling, the core conclusions regarding population structure and differentiation are strongly supported by this concordance of evidence.
I’ll, however, explicitly note this consideration and will certainly incorporate the suggested temporal coalescent frameworks in my future studies.
Comment 5. The utilized data set for phylogeographic analysis in this study is highly unbalanced. This would result in an inaccurate reconstruction of the origin of the sampled WmCSV diversity, as the phylogenetic reconstruction would tend to identify the most sampled location as the most probable state in the root of the tree. This is the case here, as well the inferred migration pathways.
Response 5: Author acknowledges the reviewer's valid theoretical concern regarding sampling imbalance/bias. However, empirical results counter this expectation: despite most sequences coming from Palestine/Israel, the phylogeographic model robustly inferred distinct origins for WmCSV DNA-A (USA) and DNA-B (Saudi Arabia). This demonstrates that the evolutionary signal was strong enough to overcome potential sampling bias, and I am confident in the inferred origins.
Howeve to test this hypothesis, I repeated the phylogeographic analysis using a reduced dataset of 60 WmA sequences (Figure S1 in the revised script) to ensure a more balanced representation from each region, primarily by reducing the number of sequences from Jordan, Palestine, and Israel. The results remained consistent with the original findings. For instance, key monophyletic clades, such as the one containing the US (NC003708) and Sudanese (KY488568) isolates, were maintained. While some minor topological rearrangements occurred—such as a Saudi isolate (KM066100) clustering with a Palestinian isolate, and isolates from Oman and Iran grouping together—they did not alter the overall conclusions (Please see Figure S1 in the revised script). These updates have been incorporated into the Methods (L119-125), Results (L263-265), and Discussion (L578-583).
Minor concern
Page 3 of 23
Comment 6. Subsections 2.2 (L109-129) and 2.4 (L152-163) should be combined. Both sections report results (MRCA and substitution rates) derived from the same Bayesian phylogenetic analysis. Presenting them together would improve the manuscript's logical flow and avoid unnecessary fragmentation.
Response 6: Author agreed and acknowledge this suggestion. Both the sub-sections have been merged. Please see subsection 2.2 in the revised script.
Comment 7. What method was used for the selection of molecular clock model and tree prior? Ideally, this would be marginal likelihoods via path sampling/stepping-stone sampling. Please clarify.
Response 7: Author employed the HKY nucleotide substitution model with estimated base frequencies to account for base substitution heterogeneity. A strict molecular clock and the Coalescent expansion growth model were used as priors to infer temporal changes in effective population size. These methodological details have been clearly specified in the revised manuscript (see Line 125-128).
Comment 8. L166-171 There is a confusion in the paragraph between natural selection and demographic history inference. The analyses performed are for natural selection analysis, not for inferring demographic history. The methods used—mismatch distribution and neutrality tests (Tajima's D and Fu's Fs)—are designed to detect population expansion under a neutral model. However, they do not directly reconstruct historical population size changes. To explicitly infer demographic history, a dedicated approach such as the Bayesian Skyline Plot (BSP) is required.
Response 8: Thanks for noting this important oversight. Its actually, natural selection inference, so the title of sub-section 2.4 (previously 2.5) has been changed to “Natural Selection Inference.” Please see L178 in the revised script.
Reviewer 3 Report
Comments and Suggestions for Authors
Watermelon chlorotic stunt virus (WmCSV), a bipartite begamovirus, poses a severe threat to global cucurbit production. In this study the authors investigate the genetic diversity, evolutionary dynamics and global dispersal of its two genomic components. They find that DNA-A (WmA) and DNA-B (WmB) strikingly contrast between each other. Together, these findings reveal evolutionary forces driving WmCSV diversification and provide critical insights into origins and ongoing global emergence of the virus.
Introduction is adequate to the subject. Materials and Methods present different computer programs/algorithms used during the analyzes and I hope other reviewers will be able to assess their applicability. Results present individual observations: (i) distinct geographic patterns and spread, (ii) evolutionary time; (iii) phylogeny (different for A vs B); (iv) genetic diversity (A vs B); recombination (A vs B). In Discussion the author summarizes the aspects of WmCSV assessment, especially the profound evolutionary asymmetry between A and B (role of spread across different hosts and recombination) and distinct origins for two genomic components. In Conclusions the results depict distinct evolution for A vs B, where A is more stable while B is a dynamic module.
In summary, I like this paper and it should be published provided that other reviewers agree with the used bioinformatics analytical tools.
One comment: the list of cited references should be shortened in their number.
Comments on the Quality of English LanguageEnglish is OK but requires correction of sporadic misspellings.
Author Response
Comment 1: the list of cited references should be shortened in their number.
Response 1: Author sincerely thanks the reviewer for his encouraging remarks. A thorough review of the cited literature was conducted to ensure relevance and accuracy. While all included references were found to be highly pertinent to the study, one or two citations that were incorrectly referenced have been identified and removed in the revised version.
Reviewer 4 Report
Comments and Suggestions for Authors
Remark:
Figure 3: WmA Tree - Is the author sure that the node with support 35 is reliable? This node clearly "falls out" from the general series. In any case, such low support deserves to be discussed
Editorial notes:
Line 43: The first mention of whitefly in the text should be the full Latin name (Bemisia tabaci). See Line 49 for comparison
Line 49: It is desirable to use a uniform font for Latin names – either regular or italics. See also Line 43
Lines 306, 362: Extra space(s)
Author Response
Comment 1. Figure 3: WmA Tree - Is the author sure that the node with support 35 is reliable? This node clearly "falls out" from the general series. In any case, such low support deserves to be discussed
Response 1: Author acknowledges that the node in the tree showing a posterior support value of 35 is indeed relatively low compared to other nodes. This low support likely reflects limited phylogenetic signal or sequence heterogeneity within that particular lineage, possibly due to geographic admixture among isolates. Author has mentioned the same in the revised script, please see Line 253-257.
Editorial notes:
Comment 2. Line 43: The first mention of whitefly in the text should be the full Latin name (Bemisia tabaci). See Line 49 for comparison
Response 2: Thank you for pointing this out. I have corrected the oversight by adding the full name (Line 43).
Comment 3. Line 49: It is desirable to use a uniform font for Latin names – either regular or italics. See also Line 43
Response 3: Citrullus lanatus has been italicized (Line 49).
Comment 4. Lines 306, 362: Extra space(s)
Response 4: The author thoroughly reviewed the manuscript for unnecessary spaces throughout the text. In addition to those kindly noted by the reviewer, several others were also identified and removed to ensure consistency and formatting accuracy.
Round 2
Reviewer 2 Report
Comments and Suggestions for Authors
The revised manuscript shows significant improvement over the original submission, and the authors have adequately addressed most of the concerns raised in the previous review. However, I have a few minor comments below for further consideration.
As previously noted, recombination screening should be conducted prior to phylogeographic and selection analyses. Therefore, I recommend that subsection 2.5 (Recombination Analysis) in the Materials and Methods be relocated to follow subsection 2.1 (Sequence Retrieval and Multiple Sequence Alignment).